# Amyloid Goiter in Familial Mediterranean Fever: Description of 42 Cases from a French Cohort and from Literature Review

**DOI:** 10.3390/jcm10091983

**Published:** 2021-05-05

**Authors:** Hélène Vergneault, Alexandre Terré, David Buob, Camille Buffet, Anael Dumont, Samuel Ardois, Léa Savey, Agathe Pardon, Pierre-Antoine Michel, Jean-Jacques Boffa, Gilles Grateau, Sophie Georgin-Lavialle

**Affiliations:** 1Internal Medicine Department and National Reference Center for Autoinflammatory Diseases and Inflammatory Amyloidosis (CEREMAIA), APHP, Tenon Hospital, Sorbonne University, 4 rue de la Chine, 75020 Paris, France; helene.vglt@gmail.com (H.V.); alexandre.terre.at@gmail.com (A.T.); lea.savey@aphp.fr (L.S.); gilles.grateau@aphp.fr (G.G.); 2Department of Pathology, APHP, Tenon Hospital, Sorbonne University, 4 rue de la Chine, 75020 Paris, France; david.buob@aphp.fr; 3Thyroid Pathologies and Endocrine Tumor Department, APHP, Pitié-Salpêtrière Hospital, Sorbonne University, 47-83 Boulevard de l’Hôpital, 75013 Paris, France; camille.buffet@aphp.fr; 4Department of Internal Medicine, Caen University Hospital, Avenue de la Côte de Nacre, 14000 Caen, France; dumont-an@chu-caen.fr; 5Department of Internal Medecine, Rennes Medical University, 2 rue Henri le Guilloux, 35000 Rennes, France; samuel.ardois@gmail.com; 6Dialysis Center, CH Sud Francilien, 40 Avenue Serge Dassault, 91100 Corbeil-Essonnes, France; agathe.pardon@chsf.fr; 7Department of Nephrology, APHP, Tenon Hospital, 4 rue de la Chine, 75020 Paris, France; pierre-antoine.michel@aphp.fr (P.-A.M.); jean-jacques.boffa@aphp.fr (J.-J.B.)

**Keywords:** familial Mediterranean fever, AA amyloidosis, goiter

## Abstract

Our aim was to describe the main features of amyloid goiter in adults with amyloidosis secondary to familial Mediterranean fever. Therefore, we analyzed cases from a French cohort of familial Mediterranean fever patients with amyloidosis and from literature review. Forty-two cases were identified: 9 from the French cohort and 33 from literature review. Ninety percent of patients were on hemodialysis for renal amyloidosis before the development of goiter. The goiter grew up rapidly in 88% of cases; 75.6% of patients were euthyroid, 58% displayed dyspnea, and 44.8% dysphagia. Various features were seen on ultrasound, from diffuse to multinodular goiter. When it was performed, fine-needle aspiration biopsy almost always revealed amyloidosis. Thirty-one patients underwent thyroidectomy: to manage compressive symptoms (72%) or rule out malignancy (27%). Histology showed mature adipose tissue in 64% of cases and lymphocytic infiltration in 21.4%. In conclusion, amyloid goiter in familial Mediterranean fever preferentially occurs in patients with end stage renal failure. Fine-needle aspiration biopsy seems to be a sensitive exam for diagnosis, but thyroidectomy remains sometimes necessary to rule out malignancy or release compressive symptoms.

## 1. Introduction

Familial Mediterranean fever (FMF) is the most frequent monogenic auto-inflammatory disorder that affects patients from the Mediterranean littoral [1]. This recessively inherited condition is caused by mutations in *MEFV* gene mostly located in exon 10. Patients suffer from episodic febrile attacks of abdominal or chest pain due to inflammation of serosal tissues. These symptoms are accompanied by biological inflammatory markers elevation and can be prevent by a daily colchicine intake. AA amyloidosis is a severe complication of FMF. This condition is characterized by fibrillar depositions of serum amyloid A (SAA) protein, especially in kidneys [2], due to chronic inflammation and particularly seen in patients with an uncontrolled disease.

The term goiter refers to an abnormal thyroid enlargement commonly due to auto-immune disease, nodules, or iodine deficiency. Goiter can be suspected on physical examination and is confirmed by ultrasonographic measurement of gland volume. Depending on the etiology, goiter can be diffuse or nodular, and thyroid hormone production can be normal, increased, or reduced.

Microscopic amyloid infiltration of the thyroid gland was first described by Rokitansky in 1855 [3] and according an autopsy cohort, it would occur in about 59% of patients suffering from AA amyloidosis [4]. The term of amyloid goiter was first used by Eiselberg in 1904 to describe the enlargement of the gland by amyloid deposits, which is far less frequent [5].

Our aim was to describe the main features of amyloid goiter in FMF patients with AA amyloidosis through the French cases from the national reference center for FMF and AA amyloidosis and a literature review.

## 2. Materials and Methods

A retrospective review of the 59 patients with AA amyloidosis secondary to FMF followed in Tenon Hospital (Paris, France) between January 1982 and January 2020 was performed.

For the literature review, a PUBMEB search was performed until August 2019 looking for case series or reports of patients. The following search was done: «amyloid goiter», «familial Mediterranean fever AND amyloid goiter» and «familial Mediterranean fever AND goiter». Articles whom abstract or full-text if needed were not available were excluded, as well as articles written in a language other than French or English, articles with no clinical details of goiter or uncertain etiology, articles dealing with non-AA-amyloidosis, and articles dealing with amyloidosis secondary to an etiology other than FMF (Figure 1).

Patients were considered as having amyloid goiter if they had a visible and/or palpable goiter or ultrasonographic enlargement (>18 cm^3^ in women, >20 cm^3^ in men) of the gland and histological or cytological exam of a thyroid sample showing positive Congo-red stained deposits with typical green-yellow birefringence. If no thyroid sample was available, the diagnosis of amyloid goiter was retained in patient with previous diagnosed AA amyloidosis if there was no evidence of alternative diagnosis, specifically: absence of auto-immune thyroiditis defined by the absence of antibodies anti-thyroid peroxidase, anti-thyroglobulin and anti-thyroid-stimulating hormone receptor (TSHR), lack of suspicion of malignant condition such as homogenous goiter, absence of nodule, and no evidence of drug-induced goiter.

The following information were collected through a standard form: age at FMF diagnosis, age at amyloidosis and goiter diagnosis, kidney function, hemodialysis, thyroid function, goiter clinical features, histological characteristics, fine-needle aspiration biopsy (FNAB), and goiter management. Euthyroidism was defined by normal levels of TSH, T3, and T4, and subclinical hypothyroidism was defined by increased TSH despite normal T3 and T4. Patients with thyroid dysfunction were screened for anti-thyroid peroxidase, anti-thyroglobulin, and anti-TSH receptor antibodies.

Data on local cases and literature cases were pooled together for the analysis. Descriptive statistics were performed through Excel software.

This observational study was based on data extracted from the JIR-cohort, an international multicenter data repository approved by the National Commission on Informatics and Liberty (CNIL authorization number No: 914677). French patients consented to be included in the JIR-cohort and were informed that data collected in medical records might be used for research study in accordance with privacy rules. Written informed consent for publication of their clinical images was obtained from the patients.

## 3. Results

Nine French cases of amyloid goiter were identified among our 59 patients with amyloidosis secondary to FMF, giving a prevalence of 15.2%. In literature, 199 articles were initially identified on PUBMED. After exclusion of irrelevant article (Figure 1), 17 articles were included. Overall, 33 cases were identified [5,6,7,8,9,10,11,12,13,14,15,16,17,18,19,20,21]. Table 1 shows the main features of the 42 patients, of whom, 24 were males (57%), mainly from Turkey (54.7%), with an average age of 30 years at diagnosis of goiter. Thirty-six patients had a pathological based diagnosis (85.7%): 22 by FNAB (61.1%), the remaining 14 by surgical sample (38.8%). For cases from the French cohort, immunohistochemistry was performed to confirm AA amyloidosis. The use of immunohistochemistry was not specified for 13 cases from the literature review. However, the case reports specified that previous biopsies of other organs revealed “AA amyloidosis” for three of them or “secondary amyloidosis” for the other ten. Microscopic proof of thyroid amyloid was not available in six patients; the diagnosis was retained, after elimination of alternative diagnosis, because patients had AA amyloidosis proven by a biopsy of another organ. Except one patient who had an isolated symptomatic involvement of thyroid, all patients whose renal function was reported (*n* = 35), suffered from chronic renal failure. Ninety percent of them were treated by hemodialysis.

Medical histories were stereotypic (Figure 2): except for two cases, amyloidosis was already known at the time of goiter diagnosis and was revealed by kidney dysfunction, proteinuria or nephrotic syndrome. Goiter occurred shortly after patients have begun hemodialysis with a median onset time of five years, ranging from 1 to 16 years. In one case, thyroid was the first organ to be symptomatic. In another case, the patient was on chronic kidney dialysis without any etiologic diagnosis. When he came from Georgia to France, a thyroidectomy was performed because of the presence of FDG^18^-PET hypermetabolic goiter (Figure 3) and revealed the presence of AA amyloidosis. The diagnosis of FMF was made retrospectively. The temporal occurrence of organ damage is however the same as in other patients.

Late diagnosis and lack of adherence to treatment were recurrent. Nine patients among the 28 whose age at diagnosis was available were diagnosed after the age of 20 (32%) and therefore the treatment by colchicine delayed. Non-compliance to colchicine was mentioned in six cases (14%).

Goiter were described as “rapidly growing” over few months in 88% of cases and could be responsible for compressive symptoms (Figure 4). Dyspnea and dysphagia were frequent (58% and 44.8%, respectively).

Endocrinological function of the gland was generally unaffected (Table 1). Euthyroidism represented 75.6% of patients. Only nine patients (24.3%) had altered thyroid function, among the 37 with available hormonal testing, including four patients (10.8%) with subclinical hypothyroidism, probably overestimated by the fact that the diagnosis was based on a single TSH, which is a variable assay prone to errors. One patient with proven amyloidosis on surgical specimen presented a subacute thyroiditis syndrome with increased serum antithyroglobulin antibody.

Ultrasonographic examination of the thyroid was available for 18 patients (43%). Images were very varied. In about half of cases (*n* = 7), the gland was hyperechoic or heterogeneous and one third of goiter were multinodular (*n* = 6). However, in some cases, the echogenicity was normal, and the goiter was diffuse and homogeneous. I^123^-scintigraphy data were rarely available (*n* = 4). There were two cases of inactive nodule, one case of diffuse hyperfixation and one case of irregular hyperfixation. All these patients were euthyroid. There was also one case with diffuse FDG uptake of the thyroid gland on the PET-CT (Figure 3). Antibodies anti-thyroid peroxidase, anti-thyroglobulin, and anti-TSHR were negative and amyloidosis was diagnosed on surgical sample of total thyroidectomy. Three cases of greasy infiltration on CT were reported (Figure 5).

When it was carried out, FNAB was a very cost-effective exam revealing amyloidosis in 88% of cases. In only 3 out of 25 cases, it showed no amyloid deposits but follicular cells, squamous metaplasia and in one case of homogenous goiter it was evocative of malignancy. Evaluation according to Bethesda system was not available.

Among the 31 patients for whom the data was available, 64% percent of patients (*n* = 20) underwent surgical procedure, 29% (*n* = 9) of them had partial thyroidectomy. Thyroidectomy was carried out in 72% of cases (*n* = 13) to manage the compressive symptoms and in 27% of cases (*n* = 5) because malignancy was suspected. In two cases, the intervention was motivated by aesthetic purpose. Histological features were detailed in 14 cases: amorphous extracellular deposits in the stroma (Figure 6) and in the wall of blood vessels were described in all cases. In two cases (14.2%), deposits were in parafollicular areas, causing displacement and compression of the thyroid follicles. Areas of mature adipose tissue were described in nine cases (64%) and lymphocytic infiltration was associated in three cases (21.4%).

## 4. Discussion

A prevalence of amyloid goiter of 15.2% was found among our French cohort of AA amyloidosis secondary to FMF, probably underestimated by the retrospective design of the study. Previous studies identified through systematic screening a prevalence of 42 to 45% of amyloid goiter among patients on chronic hemodialysis program for AA amyloid nephropathy secondary to FMF [5,16]. However, the exact prevalence remains to be refined Indeed, Altiparmak et al. [6] described only three cases of amyloid goiter among their 1100 FMF patients (0.27%), including 84 with secondary amyloidosis (3.5%).

A high proportion of rapidly growing goiter in a few months and symptomatic patients are reported here, but it could be due to a reporting bias [22]. Many patients developed a progressive asymptomatic neck enlargement a long time ago before compressive symptoms. In our center, only 55% patients had dyspnea or dysphagia (*n* = 5).

A close link seems to exist between amyloid goiter and renal failure. Twenty-eight of the 31 patients whom renal function was available were on hemodialysis (90%), which is higher than in the whole cohort of AA amyloidosis secondary to FMF followed in our center (37% of current dialysis, 54% including patients weaned from dialysis through transplantation). Few studies have reported that hypothyroidism associated with goiter or nodules would be more prevalent in patients with end-stage renal disease on hemodialysis from any causes. It has been suggested that it could be due to the Wolff-Chaikoff effect secondary to the lack of iodine excretion, to the accumulation of goitrogenic substance and to the reduction of free T3 induced by chronic dialysis [23,24,25]. However, the pathogenesis of amyloid goiter remains unclear. Although there is no data concerning the incidence of renal failure and hemodialysis on SAA levels, it has been reported that concentration of Serum Amyloid P, a protein that plays a significant part in the formation of amyloid deposits, is significantly higher in patients with chronic renal failure or on hemodialysis [26]. But kidney failure does not appear to worsen systemic AA amyloidosis deposits, so goiter expansion and thyroid dysfunction could be promoted by renal involvement. The increase in the volume of the thyroid would reveal the presence of previously asymptomatic amyloid deposits, which are very frequent in the case of systemic AA amyloidosis. There are no figures from the literature review, but in many case reports, amyloid goiter secondary to other inflammatory diseases is commonly associated with renal involvement. This association seems less marked for amyloid goiter due to AL amyloidosis. Systemic inflammation may play a role, unfortunately data for this feature were not available. It would be interesting to assess whether goiter occurs more often in patients with uncontrolled inflammatory disease. Moreover, it should be noticed that hemodialysis itself is responsible for systemic inflammation of multifactorial cause [27]. On the other hand, the link between end-stage renal failure and amyloid goiter could be non-causal and just highlight an advanced stage of disease.

Amyloid goiter is typically described as an euthyroid goiter. However, cases of hyper- or hypothyroidism have been reported [5,10,17]. A retrospective study on 11 cases of amyloid goiter found thyroid dysfunction in 90% of cases, but half of them had anti-thyroid autoantibodies [28]. Because of the prevalence of Hashimoto’s disease in the general population, thyroid tests must be interpreted with caution and patients must be tested for autoantibodies. Otherwise, chronic kidney failure is associated with abnormalities of thyroid hormones by reduction of T3 conversion from T4, urinary losses of binding protein in nephrotic syndrome and many other mechanisms [29].

Various imaging patterns can be seen depending on the amount of fat and amyloid deposition [30] and probably depending on length evolution of goiter. Fatty infiltration is responsible for increased echogenicity and decreased density in CT, whereas hypoechogenic nodules could be seen in case of predominance of amyloid deposits. Ozdemir et al. [5] suggested that at an early stage the goiter is rather homogenous and latter rather nodular.

Formal diagnosis requires histologic analysis, but FNAB seems to be a safe and sensitive method for detecting amyloid infiltration. It could avoid unnecessary and risky surgery due to the tissue fragility and to the usual complications of thyroidectomy. Of the 38 cases of amyloid goiter evaluated by Ozdemir et al., 92% had amyloidosis on thyroid aspiration [31]. However, physicians must pay attention to cells accompanying amyloid because medullary thyroid carcinoma can be associated with deposits of calcitonin amyloidosis, which can be misleading [32,33]. Moreover, diffuse and massive fatty infiltration is a consistent characteristic of amyloid goiter, reported in about two-third of cases, although it is not specific. This could be due to a metaplastic process related to hypoxia caused by amyloid deposits, leading the fibroblastic cells of the interstitium to change into mature fat cells [5]. Of note, no carcinoma was found after thyroidectomy in this FMF population; thus, in absence of compressive symptoms, it seems reasonable to avoid surgery if FNAB shows amyloid deposits.

## 5. Conclusions

Amyloid goiter in FMF patients is characterized by variable clinical presentations, but would occur preferentially in patients with end-stage renal failure secondary to amyloidosis. Thyroid function is generally preserved, so the goiter is revealed by asymptomatic neck enlargement or by compressive symptoms. FNAB seems to be a sensitive exam to identify amyloid deposits and avoid surgery in some cases.

## Figures and Tables

**Figure 1 jcm-10-01983-f001:**
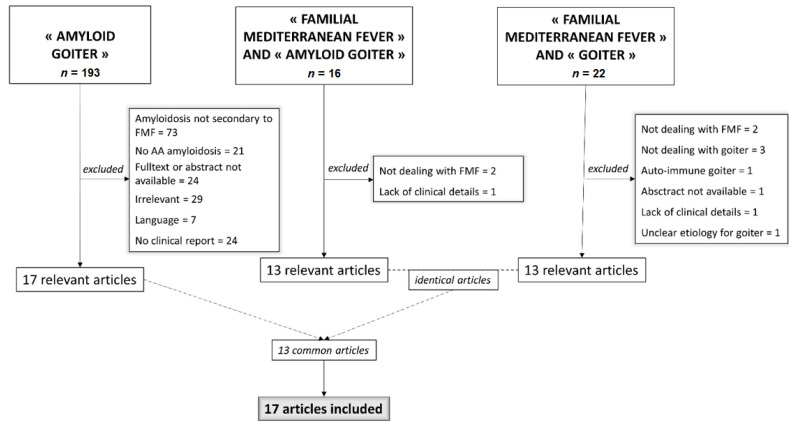
Flowchart.

**Figure 2 jcm-10-01983-f002:**
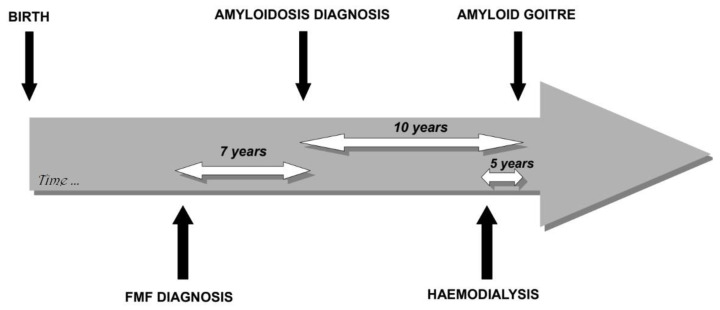
Proposed timeline for the development of amyloid goiter. The white arrows represent the median duration, in years, between each medical event indicated by a black arrow.

**Figure 3 jcm-10-01983-f003:**
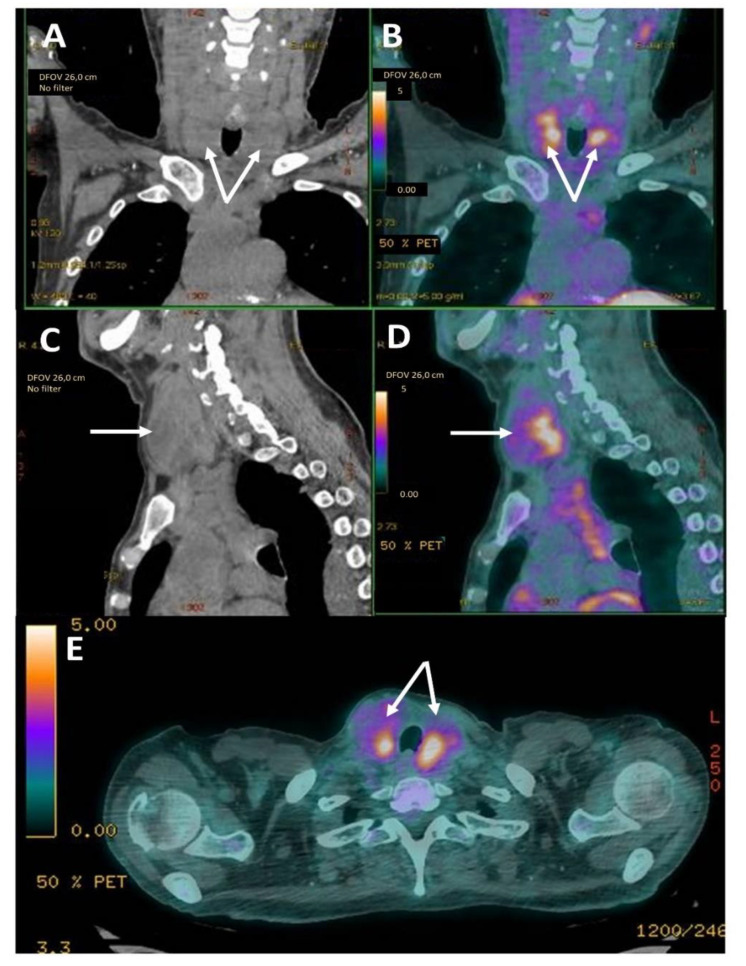
FDG^18^-PET hypermetabolic goiter. (**A**,**B**) Front section of CT, (**C**,**D**) sagittal section and (**E**) axial section of computerized tomography and FDG^18^-PET from patient B, showing enlargement and hypermetabolism of both lobes of thyroid gland (white arrows).

**Figure 4 jcm-10-01983-f004:**
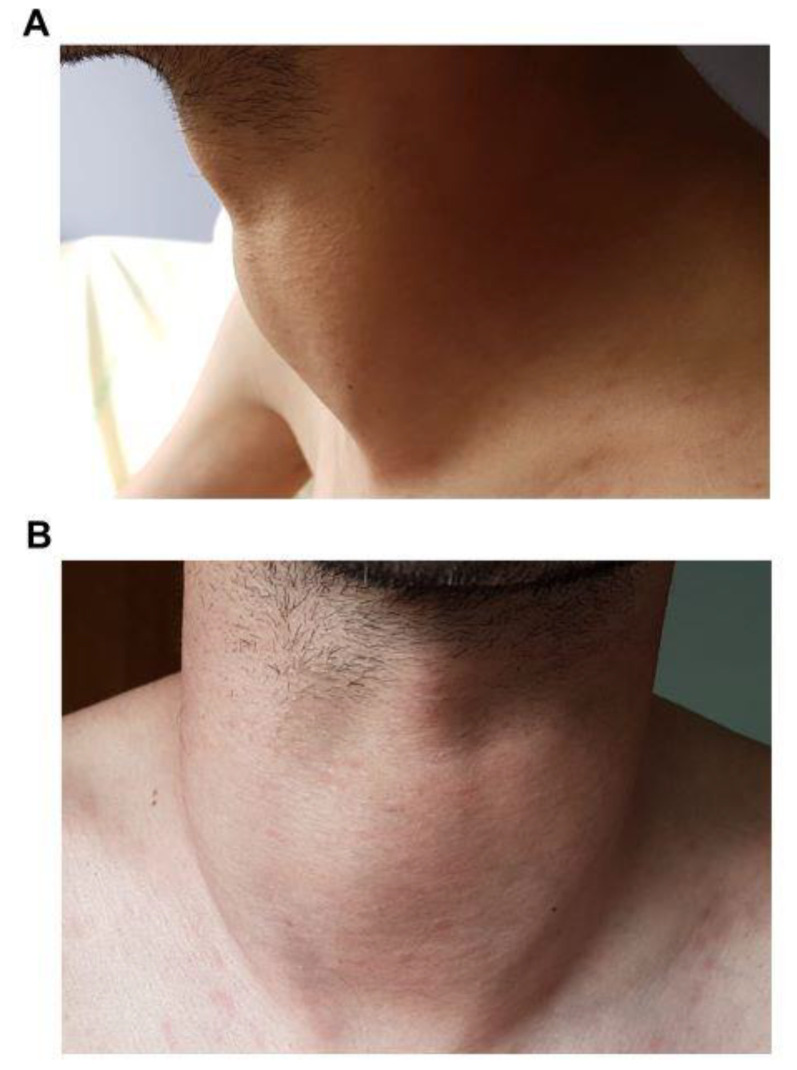
Clinical goiter.

**Figure 5 jcm-10-01983-f005:**
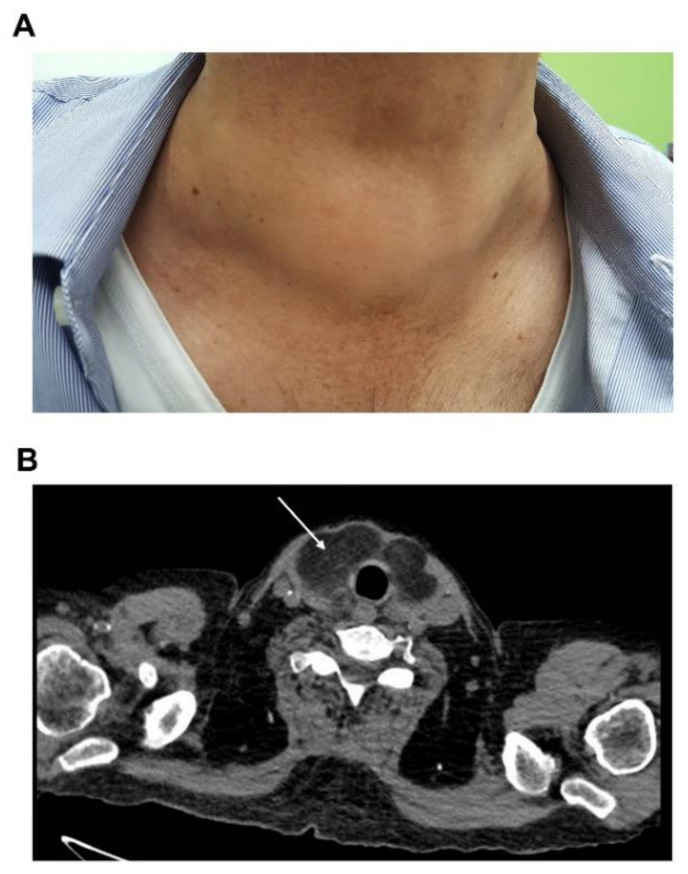
Greasy infiltration of goiter on computerized tomography. (**A**) Clinical goiter of patient C. (**B**) Cross-sectional image on TDM of patient’s C goiter showing greasy infiltration of the thyroid gland characterized by hypodense tissue (white arrow).

**Figure 6 jcm-10-01983-f006:**
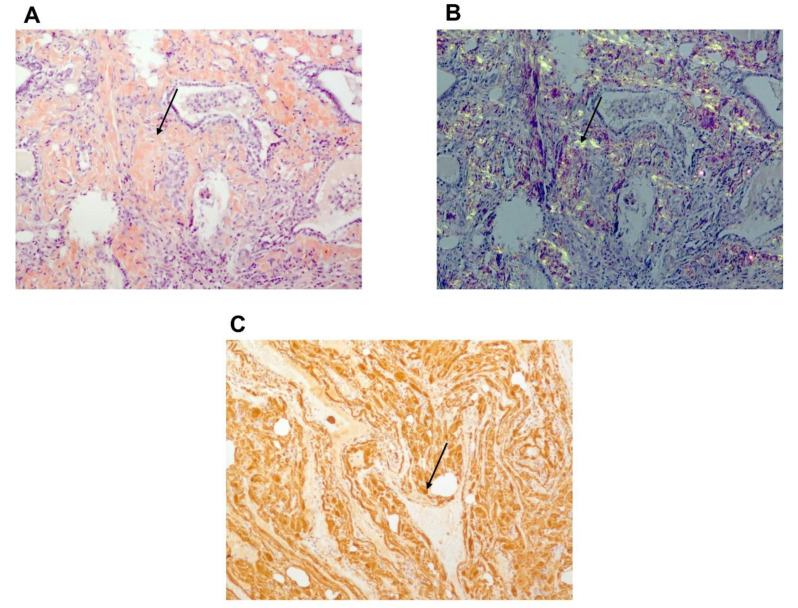
Histology: thyroid pathologic sample of patient B with amyloid deposits. (**A**) Congo red staining showing amyloid deposits. (**B**) Typical yellow-green bi-refringence of amyloid deposits under polarized light showing amyloid deposits. (**C**) Immuno-histochemistry. Brown staining due to strong fixation of anti-SAA antibody by amyloid deposits. Black narrow indicates amyloid deposit.

**Table 1 jcm-10-01983-t001:** Main features of the population.

	Patients with FMF and Amyloid Goiter
Males (*n* = 42), *n* (%)	24 (57)
**Ethnicity** (*n* = 42)	
Turkey (*n*)	23
Israel (*n*)	5
Iran (*n*)	1
Georgia (*n*)	1
Armenia (*n*)	1
Unknow (*n*)	11
**Disease characteristics**	
Age at diagnosis of FMF (*n* = 28), median (IQR)	10 (8.75–24.75)
Age at amyloidosis diagnosis (*n* = 26), median (IQR)	23.5 (15.25–36.75)
Renal failure (*n* = 35), *n* (%)	34 (97.10)
Hemodialysis (*n* = 31), *n* (%)	28 (90)
Kidney transplant (*n* = 32), *n* (%)	9 (28)
**Goiter characteristics**	
Age at diagnosis of goiter (*n* = 34), median (IQR)	30 (23–45)
Symptoms (*n* = 29)	
Asymptomatic, *n* (%)	8 (27)
Dyspnea, *n* (%)	17 (58)
Dysphagia, *n* (%)	13 (44.8)
Dyspnea and dysphagia, *n* (%)	9 (31)
Pain, *n* (%)	2 (6.8)
Unknow, *n*	13
**Thyroid function (*n* = 37)**	
Euthyroidism, *n* (%)	28 (75.6)
Hypothyroidism, *n* (%)	2 (5.4)
Subclinical hypothyroidism, *n* (%)	4 (10.8)
Hyperthyroidism, *n* (%)	2 (5.4)
Dysthyroidism with no precision, *n* (%)	1 (2.7)

## Data Availability

The data presented in this study are available on request from the corresponding author.

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
