# Peer review of "Amyloid Goiter in Familial Mediterranean Fever: Description of 42 Cases from a French Cohort and from Literature Review"

_jcm, 2021, doi:10.3390/jcm10091983_

Round 1
Reviewer 1 Report
The authors described clinicopathological features of amyloid goiter in 42 patients with familial Mediterranean fever. Nine of 42 patients were followed in Tenon Hospital, Paris, France, whereas the other 33 were from literature review. Amyloid goiter tended to occur in patents with end stage renal failure.
This is an interesting study detailing the clinical and pathological aspects of amyloid goiter associated with familial Mediterranean fever. It will be a representative study of this disease. The manuscript is well written, and I do not have any critical comments.
Minor issues and suggestions to strengthen this manuscript are raised as follows:
- “Familial Mediterranean Fever” in the abstract would be “familial Mediterranean fever”.
- AA amyloidosis is expected in patients with familial Mediterranean fever. Was this issue verified by immunohistochemistry?
- Letters in Figure 1 is too small to read. I would suggest enlarging these letters. Alternatively, please provide a higher resolution figure.
Author Response
Point 1. “Familial Mediterranean Fever” in the abstract would be “familial Mediterranean fever”.
Response 1 : We thank the reviewer for this remark. The manuscript as been corrected as requested (lines 27, 28 and 36)
Point 2. AA amyloidosis is expected in patients with familial Mediterranean fever. Was this issue verified by immunohistochemistry?
Response 2 : For cases from the French cohort, the amyloidosis subtype was proven by immunohistochemistry. The use of immunohistochemistry was unfortunetaly not precised for 13 cases from literature review. However, it was specified that previous biopsies of other organs revealed "AA amyloidosis" for 3 of them or "secondary amyloidosis" for the other 10. We nevertheless admit that this defect could generate some biais.
Point 3. Letters in Figure 1 is too small to read. I would suggest enlarging these letters. Alternatively, please provide a higher resolution figure.
Response 3 : We thank the reviewer for this remark. Higher resolution figure has been provided.
Reviewer 2 Report
This paper reports on the features of the so called amyloid goiter in a discrete cohort of patients with amyloidosis secondary to Familial Mediterranean Fever. The study cohort is enriched with the cases recruited from the available literature. Clinical and histological findings are extensively examined.
This study is undoubtedly of interest to a readership of clinicians involved in internal medicine especially for the unexpected prevalence of this manifestation in patients usually evaluated for other organ involvement. The even higher incidence of amyloid goiter in patients with ESKD in renal replacement therapy remains to be elucidated, but this observation maintains a special interest.
I have no specific criticisms about this manuscript. Clinical and histological date are soundly reported and traced back to real life and the practical clinics.
Author Response
We thank the reviewer for his consideration to this work.